# Derivation and Validation of a Clinical Prediction Score to Identify the Isolation of *Pseudomonas* in Pneumonia

Yana Maskov,[a] James M. Sanders,[a,b] Belen Tilahun,[a] Sara A. Hennessy,[c] Joan Reisch,[d,e] Meagan Johns[a]

[a]Department of Pharmacy, University of Texas Southwestern Medical Center, Dallas, Texas, USA

[b]Division of Infectious Diseases and Geographic Medicine, University of Texas Southwestern Medical Center, Dallas, Texas, USA

[c]Department of Surgery, University of Texas Southwestern Medical Center, Dallas, Texas, USA

[d]Department of Population and Data Sciences (Biostatistics), University of Texas Southwestern Medical Center, Dallas, Texas, USA

[e]Department of Family and Community Medicine, University of Texas Southwestern Medical Center, Dallas, Texas, USA

**ABSTRACT** Given the focus of existing clinical prediction scores on identifying drug-resistant pathogens as a whole, the application to individual pathogens and other institutions may yield weaker performance. This study aimed to develop a locally derived clinical prediction model for *Pseudomonas*-mediated pneumonia. This retrospective study included patients ≥18 years of age who were admitted to an academic medical center between 1 July 2010 and 31 July 2020 with a CDC National Healthcare Safety Network confirmed pneumonia diagnosis and were receiving antimicrobials during the index encounter, with a positive respiratory culture. Cystic fibrosis patients were excluded. Logistic regression analysis identified risk factors associated with the isolation of *Pseudomonas aeruginosa* from respiratory cultures within the derivation cohort ($n = 186$), which were weighted to generate a prediction score that was applied to the derivation and internal validation ($n = 95$) cohorts. A total of 281 patients met the inclusion criteria. Five predictor variables were identified, namely, tracheostomy status (4 points), chronic obstructive pulmonary disease (5 points), enteral nutrition (9 points), chronic steroid use (11 points), and *Pseudomonas aeruginosa* isolation from any culture in the prior 6 months (14 points). At a score of >11, the prediction score demonstrated a sensitivity of 52.4% (95% confidence interval [CI], 36.4 to 68.0%) and a specificity of 84.9% (95% CI, 72.4 to 93.35%) in the validation cohort. Score accuracy was 70.5% (95% CI, 60.3 to 79.4%), and the area under the receiver operating characteristic curve (AUROC) was 0.77 (95% CI, 0.68 to 0.87) in the validation cohort. A prediction score for identifying *Pseudomonas aeruginosa* in pneumonia was derived, which may have the potential to decrease the use of broad-spectrum antibiotics. Validation with larger and external cohorts is necessary.

**IMPORTANCE** In this study, we aimed to develop a locally derived clinical prediction model for *Pseudomonas*-mediated pneumonia. Utilizing a locally validated prediction score may help direct therapeutic management and be generalizable to other clinical settings and similar populations for the selection of appropriate antimicrobial coverage when data are lacking. Our study highlights a unique patient population, including immunocompromised, structural lung disease, and transplant patients. Five predictor variables were identified, namely, tracheostomy status, chronic obstructive pulmonary disease, enteral nutrition, chronic steroid use, and *Pseudomonas aeruginosa* isolation from any culture in the prior 6 months. A prediction score for identifying *Pseudomonas aeruginosa* in pneumonia was derived, which may have the potential to decrease the use of broad-spectrum antibiotics, although validation with larger and external cohorts is necessary.

**KEYWORDS** *Pseudomonas*, pneumonia, risk factors, multidrug resistant

Address correspondence to Meagan Johns, meag0701@gmail.com.

The authors declare no conflict of interest.

Pneumonia is a leading cause of hospital admissions and death in the United States, with the CDC ranking influenza and pneumonia as the eighth leading cause of death in 2017 (1, 2). Hospital acquired and ventilator associated pneumonia (HAP/VAP) are often caused by drug-resistant pathogens (DRP), such as methicillin-resistant *Staphylococcus aureus* and *Pseudomonas aeruginosa*, and treated empirically with broad-spectrum antimicrobials (3). In comparison, treatment selection is driven by local risk factors for community-acquired pneumonia (CAP) (4).

The health care-associated pneumonia (HCAP) designation was included in the 2005 HAP/VAP guidelines and provided recommendations for broad-spectrum antimicrobials to cover for DRPs, given increased risk (5). However, the updated 2016 guidelines removed this classification and recommendation because there was an increased chance of mislabeling patients at risk (3, 6). Although guidelines for pneumonia offer treatment recommendations, there are not clear directions on how to utilize risk factors to identify specific pathogens, potentially leading to overutilization of broad-spectrum antimicrobials.

Failure to initiate antibiotic therapy for DRPs portends a poor prognostic outcome. Therefore, broad-spectrum antimicrobials are often started for patients at an assumed high risk for resistant bacteria. However, if patients receive unnecessary broad-spectrum antimicrobial treatment, they are at risk of drug toxicity, resistance, and other negative consequences (7). For example, one study comparing the use of antipseudomonal combination therapy versus monotherapy found that combination therapy was associated with a higher 30-day mortality rate for CAP (7). In addition, each additional day of antipseudomonal $\beta$-lactam therapy has been associated with a 4% chance of developing a resistant organism (8). Therefore, it is helpful to utilize prediction models to identify the risk for DRPs, such as *Pseudomonas aeruginosa*, to optimize antimicrobial therapy.

Currently, clinical prediction models have primarily been developed for DRPs collectively and not for *Pseudomonas aeruginosa* specifically (9–12). Such algorithms identify criteria such as hospitalization in the preceding 90 days, antimicrobial use in the preceding 60 to 90 days, nursing home residence, prior positive culture for a DRP, and immunosuppression as possible risk factors for a DRP as the causative pathogen. However, despite the abundance of clinical prediction scores, the predictive value is diminished when criteria are applied to external populations outside the derivation site (13).

A retrospective study evaluating clinical prediction scores for DRPs in patients with CAP found variability in the scores. It revealed that more patients than necessary received broad-spectrum antimicrobials (13). A retrospective chart review conducted at a Veterans Affairs Hospital also found that the use of the drug resistance in pneumonia (DRIP) score led to an increase in the use of broad-spectrum antimicrobials, and the authors concluded that local validation of the score is necessary prior to its implementation (14). When applied at an academic medical center, the DRIP score also demonstrated decreased performance when utilized at a site other than its derivation location (15).

In our study, we aimed to identify local risk factors for the isolation of *Pseudomonas aeruginosa* in patients with pneumonia, to derive and validate a clinical prediction score specific to our patient population at an academic medical center. At our institution, we utilize MRSA nares PCR-based screening to discontinue anti-MRSA antimicrobial coverage based on the high negative predictive value for patients with pneumonia (16); therefore, we aimed to devise a more focused approach to guide empirical antipseudomonal therapy. Current prediction models do not focus on *Pseudomonas aeruginosa* and have been developed based on populations unique to their derivation sites; external validation and the use of such models may yield weaker performance. To this end, our goal was to create a clinical prediction score for identifying pseudomonal isolates in pneumonia patients at a quaternary care academic medical center that represents complex patient populations, including transplant, structural lung disease, and immunocompromised patients.

## RESULTS

A total of 672 patients were assessed for eligibility, 281 of whom (132 pseudomonal cases and 149 nonpseudomonal controls) were included in the study. A total of 391

patients were excluded because they did not meet the CDC surveillance pneumonia definition or because they had duplicate cases or multiple positive cultures. Table 1 displays a comparison of baseline characteristics between cases and controls for the entire cohort. Baseline characteristics that were significantly different included gender, shock (defined as hypovolemic, cardiogenic, obstructive, distributive, or combined/mixed type), tracheostomy status, invasive mechanical ventilation for ≥48 h, chronic steroid use, emphysema, bronchiolitis, acute respiratory distress syndrome (ARDS), congestive heart failure (CHF), enteral nutrition status prior to culture collection, proton pump inhibitor (PPI) use prior to admission, intravenous antimicrobial use within prior 90 days, and time from admission to index culture collection, as well as *Pseudomonas aeruginosa* on any culture within 6 months prior to admission and *Pseudomonas aeruginosa* on any respiratory source culture within 1 year prior to encounter.

The derivation cohort of 186 patients was used to develop the prediction score. Five variables were found to be significant for predicting pseudomonal isolates in pneumonia patients (Table 2), namely, tracheostomy status, chronic obstructive pulmonary disease (COPD), enteral nutrition through a feeding tube, chronic steroid use (defined as the use of prednisone at ≥20 mg [or equivalent] for >2 weeks), and *Pseudomonas aeruginosa* on any culture within 6 months prior to admission. The full regression model was as follows: $-2.495 + (1.073 \times \text{chronic steroid use}) + (0.522 \times \text{COPD}) + (0.430 \times \text{tracheostomy status}) + (0.902 \times \text{enteral nutrition through a feeding tube}) + (1.440 \times \textit{Pseudomonas aeruginosa} \text{ on any culture within 6 months prior to admission})$. The model fit was assessed using the Hosmer-Lemeshow test; the fit was excellent ($P = 0.91$). The maximum likelihood estimates were multiplied by 10 and rounded to the nearest whole digit to derive the points for the clinical prediction score. The prediction score derived was $(4 \times \text{tracheostomy}) + (5 \times \text{COPD}) + (11 \times \text{long-term steroid use}) + (9 \times \text{enteral nutrition}) + (14 \times \textit{Pseudomonas aeruginosa} \text{ on any culture within 6 months prior to admission})$, with a maximum score of 43.

The area under the receiver operating characteristic curve (AUROC) for the derivation cohort was 0.82 (95% confidence interval [CI], 0.76 to 0.88), with a score of >11 corresponding to the optimal breakpoint to differentiate between controls and cases (Table 3). The risk factor that had the most weight in the prediction score was *Pseudomonas aeruginosa* on any culture within 6 months prior to admission (odds ratio [OR], 17.83 [95% CI, 1.98 to 160.75]). The presence of this risk factor alone would cross the score threshold of >11 using the derived model. When the score cutoff value of >11 was applied to the derivation cohort, the score had a prediction accuracy of 75.8%. This is compared to the percent correctly identified by arbitrary assignment, which was only 51.6%, demonstrating the effectiveness in differentiating between cases and controls. Since the variable of isolation of *Pseudomonas aeruginosa* on any culture within 6 months prior to admission exceeded the score threshold of 11 independently and in clinical practice prior isolation alone may warrant empirical coverage, we performed the regression of the model omitting this variable. The resulting coefficients were similar and led to similar sensitivity, specificity, accuracy, and model fit values (data not shown). Therefore, the full model, with isolation of *Pseudomonas aeruginosa* on any culture within 6 months prior to admission, was utilized for derivation and validation cohorts.

Baseline characteristics were similar between the derivation and validation cohorts for all variables except hospital admission in the prior 90 days (Table 4). The score was applied to the internal validation cohort of 95 patients to determine its accuracy. With a threshold of >11, the prediction score demonstrated a sensitivity of 52.4% (95% CI, 36.4 to 68.0%) and a specificity of 84.9% (95% CI, 72.4 to 93.3%) in the validation cohort (Table 5). The AUROC was 0.77 (95% CI, 0.68 to 0.87), and 70.5% of the subjects were correctly identified using the model, compared to 55.6% if assigned by arbitrary assignment in the validation cohort.

## DISCUSSION

Our study highlights a unique patient population specific to our institution, including

**TABLE 1** Baseline characteristics for entire cohort

| Variable | Data for: | | $P^a$ |
|---|---|---|---|
| | Cases (*Pseudomonas*) (*n* = 132) | Controls (non-*Pseudomonas*) (*n* = 149) | |
| Male gender (no. [%]) | 88 (66.7) | 80 (53.7) | 0.03 |
| Age (mean [range]) (yr) | 61.7 (22–93) | 60 (21–88) | 0.36 |
| BMI (mean [range]) (kg/m$^2$) | 25.9 (12.3–57.3) | 27.2 (13.8–46.2) | 0.13 |
| Nonsmoker (no. [%]) | 74 (56.1) | 93 (62.8) | 0.25 |
| ICU admission (no. [%]) | 96 (72.7) | 114 (76.5) | 0.47 |
| Shock (no. [%])$^b$ | 67 (50.8) | 55 (36.9) | 0.02 |
| Tracheostomy (no. [%]) | 77 (58.3) | 46 (30.9) | <0.0001 |
| Invasive mechanical ventilation for ≥48 h (no. [%]) | 77 (58.3) | 71 (47.7) | 0.005 |
| Immunosuppressed (no. [%])$^c$ | 51 (38.6) | 43 (28.9) | 0.08 |
| Chronic steroid use (no. [%])$^d$ | 27 (20.5) | 4 (2.7) | <0.0001 |
| Solid organ transplant (no. [%]) | 17 (13) | 13 (8.7) | 0.26 |
| Hematopoietic cell transplant (no. [%]) | 2 (1.5) | 5 (3.4) | 0.45 |
| Comorbidities | | | |
| Charlson Comorbidity Index (median [range]) | 8.6 (1–22) | 7.6 (0–18) | 0.054 |
| Diabetes mellitus (no. [%]) | 41 (31.1) | 52 (34.9) | 0.49 |
| Lung disease | | | |
| Interstitial lung disease (no. [%]) | 18 (13.6) | 23 (15.4) | 0.67 |
| Pulmonary fibrosis (no. [%]) | 10 (7.6) | 10 (6.7) | 0.78 |
| Bronchiectasis (no. [%]) | 7 (5.3) | 11 (7.4) | 0.48 |
| Asthma (no. [%]) | 9 (6.8) | 10 (6.7) | 0.97 |
| Emphysema (no. [%]) | 44 (33.3) | 21 (14.1) | 0.0001 |
| COPD (no. [%]) | 42 (31.8) | 33 (22.1) | 0.07 |
| Bronchiolitis (no. [%]) | 18 (13.6) | 8 (5.4) | 0.02 |
| ARDS (no. [%]) | 18 (13.6) | 9 (6.0) | 0.03 |
| Cardiovascular disease | | | |
| CHF (no. [%]) | 25 (18.9) | 47 (31.5) | 0.02 |
| Renal disease | | | |
| Chronic kidney disease (no. [%]) | 46 (35.1) | 51 (34.2) | 0.88 |
| ESRD (no. [%]) | 19 (14.4) | 17 (11.4) | 0.46 |
| Liver disease | | | |
| Cirrhosis (no. [%]) | 1 (0.8) | 6 (4.0) | 0.13 |
| Portal hypertension (no. [%]) | 1 (0.8) | 3 (2.0) | 0.63 |
| Hepatitis (no. [%]) | 7 (5.3) | 17 (11.4) | 0.07 |
| Enteral nutrition via feeding tube prior to culture (no. [%]) | 84 (63.6) | 34 (22.8) | <0.0001 |
| PPI use prior to admission (no. [%]) | 58 (43.9) | 43 (28.9) | 0.009 |
| LTAC residence (no. [%]) | 21 (15.9) | 15 (10.1) | 0.14 |
| Nursing home residence (no. [%]) | 5 (3.8) | 4 (2.7) | 0.60 |
| Wound care | 76 (57.6) | 86 (57.7) | 0.98 |
| Pneumonia type | | | |
| CAP (no. [%]) | 66 (50.0) | 77 (51.7) | 0.78 |
| HAP/VAP (no. [%]) | 66 (50.0) | 72 (48.3) | |
| Hospital admission within prior 90 days (no. [%]) | 85 (64.4) | 79 (53.0) | 0.05 |
| Antimicrobial use within prior 90 days (no. [%]) | 70 (53.0) | 63 (42.3) | 0.07 |
| Intravenous antimicrobial use within prior 90 days (no. [%]) | 75 (56.8) | 54 (36.2) | 0.0006 |
| Time from admission to index culture collection (mean [range]) (days) | 11.5 (0–66) | 7.2 (0–66) | 0.01 |
| *P. aeruginosa* on any culture within 6 mo prior to admission (no. [%]) | 19 (14.4) | 2 (1.3) | <0.0001 |
| *P. aeruginosa* on any respiratory source culture within 1 yr prior to encounter (no. [%]) | 15 (11.4) | 4 (2.7) | 0.004 |

$^a$Comparisons of baseline characteristics between *Pseudomonas* pneumonia cases and non-*Pseudomonas* controls were performed by utilizing the chi-square test or Fisher's exact test for categorical variables and Student's *t* test for continuous variables. Two-sided *P* values of <0.05 were considered significant.
$^b$Shock was defined as hypovolemic, cardiogenic, obstructive, or combined/mixed type.
$^c$Immunosuppression was defined as any one of the following: neutropenia (ANC or WBC count of <500 cells/mm$^3$), leukemia/lymphoma or HIV positive with CD4$^+$ cell count of <200 cells/mm$^3$, history of splenectomy, history of solid organ transplant or hematopoietic stem cell transplant, receiving cytotoxic chemotherapy, or taking prednisone at ≥20 mg or equivalent for >2 weeks.
$^d$Chronic steroid use was defined as the use of prednisone at ≥20 mg (or equivalent) for >2 weeks.

**TABLE 2** Identified risk factors for isolation of *Pseudomonas* in pneumonia

| Risk factor | OR (95% CI) | *P* |
|---|---|---|
| Enteral nutrition prior to culture | 6.05 (2.93–12.51) | 0.000 |
| Chronic steroid use[a] | 8.55 (1.69–43.25) | 0.0095 |
| *P. aeruginosa* on any culture within 6 mo prior to admission | 17.83 (1.98–160.75) | 0.0103 |
| COPD | 2.84 (1.25–6.44) | 0.0123 |
| Tracheostomy | 2.36 (1.16–4.83) | 0.019 |

[a]Chronic steroid use was defined as the use of prednisone at ≥20 mg (or equivalent) for >2 weeks.

immunocompromised, structural lung disease, and transplant patients. Among these groups, predictors for the isolation of *Pseudomonas aeruginosa* in patients diagnosed with pneumonia were tracheostomy status, COPD, enteral nutrition through a feeding tube, chronic steroid use, and *Pseudomonas aeruginosa* on any culture within 6 months prior to admission, with the latter independently crossing the score threshold for identifying at-risk patients. When applied to both the derivation and validation cohorts, the scoring model accurately identified isolation of *Pseudomonas aeruginosa* in patients with pneumonia.

Risk factor variables initially included in our regression model were similar to those identified in previous publications for DRP prediction models, such as antimicrobial use in the prior 90 days, enteral nutrition support, gastric acid suppression, and prior infection with a DRP (9–12). However, the predictor variables incorporated into our final scoring model were unique to our specific patient population. Tracheostomy status and COPD were found to be predictive variables included in the score and reflect the large number of patients with structural lung disease at our institution. This is supported in previous studies and guidelines, which show that *Pseudomonas aeruginosa* colonization leads to an increased risk of hospitalization for COPD exacerbation, as well as increased rates of *Pseudomonas* CAP in patients with chronic lung diseases (tracheostomy, bronchiectasis, and/or severe COPD) (17–19). Chronic steroid use is also reflective of the large number of patients with structural lung disease or a history of solid organ transplantation in this cohort. Immunosuppression or the use of corticosteroids has also been associated with the isolation of DRPs in pneumonia (3, 10).

*Pseudomonas aeruginosa* on any culture within 6 months prior to admission was found to be the greatest predictor of *Pseudomonas aeruginosa* isolation in pneumonia in our model. By weighing this risk factor with a score of 14, patients growing *Pseudomonas aeruginosa* on any culture within 6 months prior to admission would automatically be identified as a case and would already pass the threshold of >11. This is similar to previous models and publications, which found that prior infection with a DRP on any culture within 1 year of admission was a risk factor for drug-resistant pneumonia (12, 15).

In a previous study comparing the use of the DRIP score to other prediction models such as the HCAP criteria, the DRIP score performed more favorably in the validation cohort, with the AUROC values of the models ranging from 0.72 to 0.88 (12). This is comparable to the AUROC of 0.77 in our validation cohort. Given that the previous study compared the different models derived from single institutions, the AUROC values of these models would likely be different if they were applied at our institution, as shown by others (15). In addition, the other models included

**TABLE 3** Sensitivity and specificity for each score threshold

| Score comparison | Sensitivity (%) | Specificity (%) | Accuracy (%) |
|---|---|---|---|
| 0 vs 4–43 | 92.2 | 35.4 | 62.9 |
| 0–4 vs 5–43 | 83.3 | 57.3 | 69.9 |
| 0–5 vs 9–43 | 80.0 | 68.8 | 74.2 |
| 0–9 vs 11–43 | 63.3 | 86.5 | 75.3 |
| 0–11 vs 13–43 | 63.3 | 87.5 | 75.8 |
| 0–13 vs 14–43 | 45.6 | 95.8 | 71.5 |
| 0–14 vs 16–43 | 40.0 | 97.9 | 69.9 |
| 0–16 vs 18–43 | 35.6 | 97.9 | 67.7 |
| 0–18 vs 19–43 | 25.6 | 99.0 | 62.9 |

**TABLE 4** Characteristics for derivation and validation cohorts

| Variables | Data for: | | |
| --- | --- | --- | --- |
| | Derivation cohort (*n* = 186) | Validation cohort (*n* = 95) | *P*ᵃ |
| Male gender (no. [%]) | 110 (59.1) | 58 (61.0) | 0.76 |
| Age (mean [range]) (yr) | 59.9 (22–92) | 62.7 (21–93) | 0.14 |
| BMI (mean [range]) (kg/m²) | 26.5 (12.3–54.1) | 26.8 (13.8–57.3) | 0.77 |
| Nonsmoker (no. [%]) | 75 (40.3) | 38 (40.4) | 0.99 |
| Tracheostomy (no. [%]) | 87 (46.8) | 36 (37.9) | 0.16 |
| Invasive mechanical ventilation for ≥48 h (no. [%]) | 97 (52.2) | 51 (53.7) | 0.96 |
| Immunosuppressed (no. [%])ᵇ | 55 (29.6) | 39 (41.1) | 0.05 |
| Chronic steroid use (no. [%])ᶜ | 20 (10.8) | 11 (11.6) | 0.83 |
| Comorbidities | | | |
| Diabetes mellitus (no. [%]) | 62 (33.3) | 31 (32.6) | 0.91 |
| Lung disease | | | |
| Emphysema (no. [%]) | 44 (23.7) | 21 (22.1) | 0.77 |
| COPD (no. [%]) | 51 (27.4) | 24 (25.3) | 0.70 |
| Bronchiolitis (no. [%]) | 17 (9.1) | 9 (9.5) | 0.93 |
| ARDS (no. [%]) | 18 (9.7) | 9 (9.5) | 0.96 |
| Cardiovascular disease | | | |
| CHF (no. [%]) | 47 (25.3) | 25 (26.3) | 0.85 |
| Kidney disease | | | |
| ESRD (no. [%]) | 22 (11.8) | 14 (14.7) | 0.49 |
| Enteral nutrition via feeding tube prior to culture (no. [%]) | 82 (44.1) | 36 (37.9) | 0.32 |
| PPI use prior to admission (no. [%]) | 74 (39.8) | 27 (28.4) | 0.06 |
| LTAC residence (no. [%]) | 27 (14.5) | 9 (9.5) | 0.23 |
| Nursing home residence (no. [%]) | 6 (3.2) | 3 (3.2) | 0.98 |
| Pneumonia type | | | |
| CAP (no. [%]) | 93 (50.0) | 50 (52.6) | 0.68 |
| HAP/VAP (no. [%]) | 93 (50.0) | 45 (47.4) | |
| Hospital admission in prior 90 days (no. [%]) | 117 (62.9) | 47 (49.5) | 0.03 |
| Antimicrobial use within prior 90 days (no. [%]) | 94 (50.5) | 39 (41.1) | 0.13 |
| Intravenous antimicrobial use within prior 90 days (no. [%]) | 91 (48.9) | 38 (40.0) | 0.16 |
| Time from admission to index culture collection (mean [range]) (days) | 8.7 (0–66) | 10.1 (0–63) | 0.43 |
| *P. aeruginosa* on any culture within 6 mo prior to admission (no. [%]) | 15 (8.1) | 6 (6.3) | 0.60 |
| *P. aeruginosa* on any respiratory source culture within 1 yr prior to encounter (no. [%]) | 16 (8.6) | 3 (3.2) | 0.09 |

ᵃComparisons of baseline characteristics between the derivation and validation cohorts were performed by utilizing the chi-square test or Fisher's exact test for categorical variables and Student's *t* test for continuous variables. Two-sided *P* values of <0.05 were considered significant.

ᵇImmunosuppression was defined as any one of the following: neutropenia (ANC or WBC count of <500 cells/mm³), leukemia/lymphoma or HIV positive with CD4⁺ cell count of <200 cells/mm³, history of splenectomy, history of solid organ transplant or hematopoietic stem cell transplant, receiving cytotoxic chemotherapy, or taking prednisone at ≥20 mg or equivalent for >2 weeks.

ᶜChronic steroid use was defined as the use of prednisone at ≥20 mg (or equivalent) for >2 weeks.

DRPs collectively or MRSA and not specifically *Pseudomonas aeruginosa*, as was done in this analysis.

There is some distinction in the literature among risk factors for the isolation of specific DRPs, such as MRSA and *Pseudomonas*, in pneumonia. In one study evaluating the risk of MRSA in pneumonia, a variety of risk factors were identified when HAP/VAP patients were excluded (20). Predictors unique for MRSA included age of <30 years or >79 years, intensive care unit (ICU) admission (at or before index culture), and comorbidities, including cerebrovascular disease prior to admission, dementia, and females with diabetes mellitus. Variability in risk factors between MRSA and *Pseudomonas aeruginosa* necessitates the need for clinical prediction scores unique to specific DRPs.

**TABLE 5** Score performance with derivation and validation cohorts

| Cohort | Sensitivity (95% CI) (%) | Specificity (95% CI) (%) | AUROC (95% CI) | Accuracy (%) (95% CI) (%) |
| --- | --- | --- | --- | --- |
| Derivation (*n* = 186) | 63.3 (52.5–73.3) | 87.5 (79.1–93.4) | 0.82 (0.76–0.88) | 75.8 (69.0–81.8) |
| Validation (*n* = 95) | 52.4 (36.4–68.0) | 84.9 (72.4–93.3) | 0.77 (0.68–0.87) | 70.5 (60.3–79.4) |

The most commonly identified pathogens in the control group in our analysis included MRSA (19%), methicillin-sensitive *Staphylococcus aureus* (MSSA) (18%), *Stenotrophomonas maltophilia* (11%), and *Klebsiella* spp. (8%). Despite the isolation of other DRPs, we were still able to identify risk factors for *Pseudomonas aeruginosa* specifically in this patient cohort. It is important to be able to stratify pathogen-specific risk, because different pathogens are associated with different risk factors and may guide antimicrobial management more appropriately.

Limitations of this study include the fact that the prediction score was validated only in a retrospective cohort that was from the same institution as that from which the score was derived. Additionally, the small sample size prevented an analysis of the different types of pneumonia (CAP versus HAP/VAP). With a larger sample size, risk factors for pseudomonal growth in pneumonia patients could be evaluated specifically for CAP, as opposed to all pneumonia types. In addition, by having a larger sample size, more predictor variables might be incorporated and evaluated for inclusion in the prediction model. Additionally, the prevalence of *Pseudomonas*-mediated pneumonia at our institution was not calculated in this analysis due to the case-control design of the study limiting generalizability. Lastly, although there were no statistically significant differences between the derivation and validation cohorts except for hospitalization in the prior 90 days, potential differences could still exist between the two cohorts.

Unnecessary broad-spectrum antimicrobial treatment increases patient risk for negative consequences. Utilizing a locally validated prediction score can help guide therapeutic management. Current clinical prediction scores have been derived for identifying DRPs. However, risk factors are distinctive for individual pathogens. With the development and internal validation of a clinical prediction score for identifying isolation of *Pseudomonas aeruginosa* in pneumonia patients at the University of Texas Southwestern Medical Center (UTSW), the application to a unique institution and its populations may facilitate choosing appropriate antimicrobial coverage when data are lacking. The score can also serve to determine the deescalation of antibiotic therapy in order to reduce costs and unintended antimicrobial consequences such as *Clostridium difficile* infection and resistance. Further validation with prospective studies and larger internal and external cohorts is needed to evaluate the use of this prediction score in identifying *Pseudomonas aeruginosa* isolation in pneumonia patients.

## MATERIALS AND METHODS

This was a single-center, retrospective cohort study that included adult patients (≥18 years of age) with pneumonia who were admitted to the UTSW Medical Center, received antimicrobials during the index encounter, and had a positive respiratory culture between 1 July 2010 and 31 July 2020. The study was approved by the UTSW Medical Center institutional review board (reference number STU-2020-1161). Informed consent was not required according to the UTSW Medical Center institutional review board national and institutional standards for a retrospective cohort study.

A microbiologic diagnosis was confirmed with a positive culture from a sputum or bronchoalveolar lavage (BAL) fluid sample, tracheal aspirate sample, or pleural fluid sample with *Pseudomonas aeruginosa* (cases) or a respiratory bacterial isolate other than *Pseudomonas aeruginosa* (controls). Duplicate cases arose from having positive cultures across multiple hospital admissions. In such cases, the hospital admission with the index positive culture was used. For inclusion, the patients had to meet the CDC surveillance definition for clinical pneumonia by exhibiting defined signs and symptoms and having two or more serial chest imaging test results with at least one of the following: infiltrate, consolidation, or cavitation. Signs and symptoms were defined as a fever (>38.0°C/100.4°F), leukopenia (≤4,000 white blood cells [WBC]/mm$^3$), or leukocytosis (≥12,000 WBC/mm$^3$) and at least two of the following: new onset or change in character of sputum or increased respiratory secretions or suctioning requirements, new onset or worsening cough or dyspnea/tachypnea, rales or bronchial breath sounds, or worsening gas exchange (21). Patients with all pneumonia types, including CAP, HAP/VAP, and HCAP, were enrolled. Patients with cystic fibrosis were excluded from this study.

During the period reviewed, 283 patients were identified who had a positive culture from a sputum or BAL fluid sample, tracheal aspirate sample, or pleural fluid sample with *Pseudomonas aeruginosa*; of those screened, 132 met the inclusion criteria and the CDC definition for clinical pneumonia. A total of 1,167 patients were identified with a positive culture from a sputum or BAL fluid sample, tracheal aspirate sample, or pleural fluid sample with a respiratory bacterial isolate other than *Pseudomonas aeruginosa*. Every third patient with a respiratory bacterial isolate other than *Pseudomonas aeruginosa* was assessed for eligibility, which yielded a total of 149 controls. A total of 281 patients met the inclusion

criteria. The cohort was randomly divided into a derivation cohort of 186 patients and a validation cohort of 95 patients utilizing SPSS.

The primary objective was the identification of risk factors associated with the isolation of *Pseudomonas aeruginosa* in pneumonia patients to develop a locally derived clinical prediction score. Comparisons of baseline characteristics between patients with pseudomonal and nonpseudomonal respiratory culture isolates were performed utilizing the chi-square test or Fisher's exact test for categorical variables and Student's *t* test for continuous variables (Table 1). Risk factor variables were selected *a priori* based on current literature and practice trends (9–12, 14).

Univariate analyses were used to determine which factors had statistically significant differences between cases and controls. A stepwise logistic regression using SAS version 9.4 (Cary, NC) was performed to determine the predictor variables. A two-sided *P* value of <0.05 was considered statistically significant and an entrance criterion for the stepwise logistic regression, with a stay criterion of 0.10. Factors included in the regression model were age, gender, body mass index (BMI), invasive mechanical ventilation status at the time of index culture collection, duration of mechanical ventilation during the index encounter and prior to index culture collection, type of pneumonia, emphysema, COPD, bronchiolitis, ARDS, chronic steroid use (prednisone at ≥20 mg [or equivalent] for >2 weeks) prior to index culture collection, immunosuppression status, diabetes mellitus, smoking status, CHF, end-stage renal disease (ESRD), hospital admission in prior 90 days, time from admission to index culture collection, antimicrobial and intravenous antimicrobial use within prior 90 days, enteral nutrition (via feeding tube) status during the index encounter and prior to index culture collection, PPI use prior to admission, long-term acute care (LTAC) residence, nursing home residence, tracheostomy status during the index encounter and prior to index culture collection and at the time of index culture collection, and *Pseudomonas aeruginosa* on any culture or any respiratory culture within 6 months prior to encounter or 1 year prior to encounter, respectively. Lung diseases, such as asthma, emphysema, COPD, and bronchiolitis, were defined as being identified on the problem list or by the International Classification of Diseases (ICD) Ninth or Tenth Revision codes in the index encounter. Immunosuppression was defined as any one of the following: neutropenia (absolute neutrophil count [ANC] or WBC count of <500 cells/mm³), leukemia/lymphoma or HIV positive with a CD4⁺ cell count of <200 cells/mm³, history of splenectomy, history of solid organ transplant or hematopoietic stem cell transplant, receiving cytotoxic chemotherapy, or taking prednisone at ≥20 mg or equivalent for >2 weeks. Five predictor variables were identified by the stepwise logistic regression to be incorporated into the score (Table 2). The maximum likelihood estimates were multiplied by 10 and rounded to the nearest whole digit to derive the points for the clinical prediction score. A receiver operating characteristic (ROC) analysis was performed, and sensitivity, specificity, and accuracy were assessed for various possible scores. The highest accuracy was for a score of 11 (Table 3). The Transparent Reporting of a multivariable prediction model for Individual Prognosis Or Diagnosis (TRIPOD) checklist was reviewed for reporting (22).

## ACKNOWLEDGMENTS

We thank Donglu Xie for her assistance in data collection.

This work was not supported by any grant or funding from funding agencies in the public, commercial, or not-for-profit sectors.

We have no conflicts of interest to declare.

Y.M., J.M.S., B.T., S.A.H., J.R., and M.J. participated in the performance of the research, research design, data analysis, and writing of the article. J.R. participated in the performance of data analysis and research design. All authors participated in the development of the final edition of the manuscript.

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
