## [Reviewer comments · Microbiology Spectrum]

Microbiology Spectrum

Derivation and validation of a clinical prediction score to identify the isolation of *Pseudomonas* in pneumonia

Yana Maskov, James Sanders, Belen Tilahun, Sara Hennessy, Joan Reisch, and Meagan Johns

Corresponding Author(s): Meagan Johns, University of Texas Southwestern Medical Center

Review Timeline:

Submission Date:	February 7, 2022
Editorial Decision:	March 8, 2022
Revision Received:	April 24, 2022
Accepted:	May 3, 2022

Editor: Bonnie Prokesch

Reviewer(s): Disclosure of reviewer identity is with reference to reviewer comments included in decision letter(s). The following individuals involved in review of your submission have agreed to reveal their identity: Tristan Timbrook (Reviewer #1)

Transaction Report:

DOI: <https://doi.org/10.1128/spectrum.00424-22>

March 8, 2022

Dr. Meagan Lee Johns
University of Texas Southwestern Medical Center
Pharmacy
Dallas, Texas

Re: Spectrum00424-22 (Derivation of a clinical prediction score to identify the isolation of Pseudomonas in pneumonia)

Dear Dr. Meagan Lee Johns:

Link Not Available

Sincerely,

Bonnie Prokesch

Journals Department
Reviewer comments:

Reviewer #1 (Comments for the Author):

Overall: The present study is a derivation and validation study of a clinical prediction model for pseudomonas as etiology of pneumonia among CAP, HAP, and VAP patients. Overall, the study found moderate discrimination performance for the classification of pseudomonas etiology, yield potential benefit in driving appropriate empiric antimicrobial therapy. These data are informative and interesting. I applaud the efforts of the authors in their work. However, there are several issues that should be addressed before publishing.

Title: Would consider indicating the validation analysis as well in the title

Page 5 Lines 118-120: May consider revising the context here of the problem leading to the objective of the study as still somewhat opaque as to why only developing for Pseudomonas. Perhaps as avoidance of MRSA therapy can be driven by guideline-supported testing of nasal surveillance for this determination? Thus practically, could drive to CAP coverage if no Pseudomonas per risk score and negative nasal swab (and presumably low CRE/ESBL setting)? Even the developers of DRIP have more or less followed this paradigm so perhaps reasonable argument (Webb BJ, Sorensen J, Mecham I, Buckel W, Ooi L, Jephson A, Dean NC. Antibiotic Use and Outcomes After Implementation of the Drug Resistance in Pneumonia Score in ED Patients With Community-Onset Pneumonia. *Chest*. 2019 Nov;156(5):843-851. doi: 10.1016/j.chest.2019.04.093. Epub 2019 May 8. PMID: 31077649.)

Page 6-7 Lines 149-178: Recommend following (and reporting following thereof) the TRIPOD checklist for studies on multivariable prediction derivation and/or validation. Similarly (and in TRIPOD checklist), please provide a sample size justification.

Moons KG, Altman DG, Reitsma JB, Ioannidis JP, Macaskill P, Steyerberg EW, Vickers AJ, Ransohoff DF, Collins GS. Transparent Reporting of a multivariable prediction model for Individual Prognosis Or Diagnosis (TRIPOD): Explanation and Elaboration. *Ann Intern Med*. 2015;162(1):W1-W73. PMID: 25560730

Page 7 Lines 159-169: For all time-varying potential predictors (e.g. mechanical ventilation), please clarify they were obtained before or at time of index culture. If they were past this time point, would exclude from model as would be unavailable for prediction till later in patient encounter.

Page 9 Lines 198-202: Again, as recommended per TRIPOD, would include the full regression model (in addition to current risk score model) including the beta coefficients and intercept.

Page 10 Lines 28-31 and Pages 12-13 Lines 284-290: How was NPV and PPV derived if this was a case control study and therefore prevalence of the pathogen is not derivable from these patients? Do you have the prevalence from other data within your system or was assumed from literature?

Page 10 Lines 228-231: "the scoring model improved the accuracy" compared to what? What was it improved from? As no comparison was made (e.g. to say clinical judgement), would consider revising verbiage. Similarly, on page 12 Lines 272-277, would avoid "prevalence" verbiage given this is a case control study.

Reviewer #2 (Comments for the Author):

The authors present a single-center retrospective study to identify their center's local risk factors for pneumonia due to Pseudomonas. They acknowledge that other risk scores have been developed but do not perform as well outside of the center at which they were derived. To this end, they sought to look at their population to determine risk factors for Pseudomonas amongst all those admitted for CAP/HAP/VAP. One strength of the study is the use of the standardized CDC definition of pneumonia and microbiologic diagnosis in all included patients. One weakness, which they acknowledge, is the heterogeneous inclusion of CAP/HCAP/VAP/HAP cases in the score.

Comments:

1. Line 99: Recommend clarifying this; rather than "4% risk of resistance" perhaps saying "4% chance of developing a resistant organism".

2. Line 144: patients selected for the control group were not selected "at random" if every 3rd non-Pseudomonas culture was selected

3. How were COPD, emphysema, bronchiolitis and asthma defined? There is a lot of overlap between these entities. Since COPD was part of the final score, would suggest adding more clarity surrounding how these were defined.

4. Most of the predictive weight in the score comes from a prior positive Pseudomonas culture. In practice, empiric Pseudomonas coverage would be given to these patients without calculating a risk score. The patients without prior culture are the ones where a prediction score would be useful. Would recommend presenting an alternative model that does NOT include this variable, along with performance characteristics.

In the introduction, the authors correctly state that a limitation of prior studies is lack of availability to other sites, as mentioned above. Here, they present a study that can be potentially useful at their own site, and rightly point out that the generalizability needs to be determined.

Staff Comments:

Preparing Revision Guidelines

Please return the manuscript within 60 days; if you cannot complete the modification within this time period, please contact me. If you do not wish to modify the manuscript and prefer to submit it to another journal, please notify me of your decision immediately so that the manuscript may be formally withdrawn from consideration by Microbiology Spectrum.

Meagan Johns, PharmD, MBA, BCCCP, BCPS, BCNSP
UT Southwestern Medical Center
6201 Harry Hines Blvd.
Dallas, TX 75390
214-633-4602
Meag0701@gmail.com

April 24, 2022

Dear Dr. Bonnie Prokesch,

Thank you for your comprehensive review of our manuscript titled “Derivation and validation of a clinical prediction score to identify the isolation of *Pseudomonas* in pneumonia”. We appreciate the thoughtful editorial and reviewer feedback. In response to the reviewer comments, we have updated our manuscript to add clarity. Please find enclosed a copy with yellow highlighted changes and a clean version of the manuscript. Our responses to reviewer comments are noted below.

Reviewer #1

Overall: The present study is a derivation and validation study of a clinical prediction model for pseudomonas as etiology of pneumonia among CAP, HAP, and VAP patients. Overall, the study found moderate discrimination performance for the classification of pseudomonas etiology, yield potential benefit in driving appropriate empiric antimicrobial therapy. These data are informative and interesting. I applaud the efforts of the authors in their work. However, there are several issues that should be addressed before publishing.

- 1.) Title: Would consider indicating the validation analysis as well in the title

Response: We updated the title to reflect the reviewer’s comment (Page 1). To reflect the intent of the reviewer’s comment, we also updated Page 5, Line 117 with “and validate”.

- 2.) Page 5 Lines 118-120: May consider revising the context here of the problem leading to the objective of the study as still somewhat opaque as to why only developing for *Pseudomonas*. Perhaps as avoidance of MRSA therapy can be driven by guideline-supported testing of nasal surveillance for this determination? Thus practically, could drive to CAP coverage if no *Pseudomonas* per risk score and negative nasal swab (and presumably low CRE/ESBL setting)? Even the developers of DRIP have more or less followed this paradigm so perhaps reasonable argument (Webb BJ, Sorensen J, Mecham I, Buckel W, Ooi L, Jephson A, Dean NC. Antibiotic Use and Outcomes After Implementation of the Drug Resistance in Pneumonia Score in ED Patients With Community-Onset Pneumonia. *Chest*. 2019

Nov;156(5):843-851. doi: 10.1016/j.chest.2019.04.093. Epub 2019 May 8. PMID: 31077649.)

Response: We agree with the reviewer that our intent for pursuing a focused approach to determine risk factors for *Pseudomonas* alone was not clearly stated in the introduction. We have modified the manuscript to reflect our approach to focus on *Pseudomonas* in tandem with alternative diagnostics (i.e., MRSA nares PCR) to guide anti-MRSA coverage with the addition of a necessary citation. Please see Page 5, Lines 118-121.

- 3.) Page 6-7 Lines 149-178: Recommend following (and reporting following thereof) the TRIPOD checklist for studies on multivariable prediction derivation and/or validation. Similarly (and in TRIPOD checklist), please provide a sample size justification.

Moons KG, Altman DG, Reitsma JB, Ioannidis JP, Macaskill P, Steyerberg EW, Vickers AJ, Ransohoff DF, Collins GS. Transparent Reporting of a multivariable prediction model for Individual Prognosis Or Diagnosis (TRIPOD): Explanation and Elaboration. *Ann Intern Med.* 2015;162(1):W1-W73. PMID: 25560730

Response: We appreciate the reviewer’s suggestion to include the TRIPOD checklist as part of the manuscript. To address specifics, please find below the checklist with our response to items that were absent (noted in red font) in the original manuscript. Please note, we did not include the checklist within the body of the manuscript. In lieu of the entire checklist, we have added a statement within the Methods section to report that we followed the checklist and added the above reviewer referenced article (Page 9, Lines 196-198).

Regarding sample size justification, we have included further details to provide transparency in how we arrived at the final sample of patients used to generate the prediction model (Page 7, Lines 148-160). In short, the final number of patients included was determined by the number of patients meeting inclusion criteria of documented *Pseudomonas aeruginosa* and CDC defined pneumonia with subsequent screening of controls to ensure a balance of cases to controls of approximately 1:1.

TRIPOD Checklist

Section/Topic	Item	Checklist	Page
Title and Abstract			
Title	1	Identify the study as developing and/or validating a multivariable prediction model, the target population, and the outcome to be predicted. Validation analysis was added to the title as recommended.	1
Abstract	2	Provide a summary of objectives, study design, setting, participants, sample size, predictors, outcome, statistical analysis, results, and conclusions.	2-3
Introduction			
Background and	3a	Explain the medical context (including whether diagnostic	4-5

Objectives		or prognostic) and rationale for developing or validating the multivariable prediction model, including references to existing models.	
	3b	Specify the objectives, including whether the study describes the development or validation of the model or both.	5
Methods			
Source of Data	4a	Describe the study design or source of data (e.g., randomized trial, cohort, or registry data), separately for the development and validation data sets, if applicable.	6-7
	4b	Specify the key study dates, including start of accrual; end of accrual; and, if applicable, end of follow-up.	6-7
Participants	5a	Specify key elements of the study setting (e.g., primary care, secondary care, general population) including number and location of centres.	6-7
	5b	Describe eligibility criteria for participants.	6-7
	5c	Give details of treatments received, if relevant.	6-7
Outcome	6a	Clearly define the outcome that is predicted by the prediction model, including how and when assessed.	8
	6b	Report any actions to blind assessment of the outcome to be predicted.	N/A
Predictors	7a	Clearly define all predictors used in developing or validating the multivariable prediction model, including how and when they were measured. Statements were added to clarify that time-varying potential predictors were obtained prior to or at the time of index culture collection as recommended.	7-8
	7b	Report any actions to blind assessment of predictors for the outcome and other predictors.	N/A
Sample Size	8	Explain how the study size was arrived at. Explanation of sample size added as recommended (see above).	6-7
Missing data	9	Describe how missing data were handled (e.g., complete-case analysis, single imputation, multiple imputation) with details of any imputation method.	N/A
Statistical analysis methods	10a	Describe how predictors were handled in the analyses.	6-8
	10b	Specify type of model, all model-building procedures (including any predictor selection), and method for internal validation.	6-10
	10c	Specify all measures used to assess model performance and, if relevant, to compare multiple models.	6-10
Risk groups	11	Provide details on how risk groups were created, if done.	N/A
Results			
Participants	13a	Describe the flow of participants through the study, including the number of participants with and without the outcome and, if applicable, a summary of the follow-up time. A diagram may be helpful.	7,9

	13b	Describe the characteristics of the participants (basic demographics, clinical features, available predictors), including the number of participants with missing data for predictors and outcome.	9,19-20
Model development	14a	Specify the number of participants and outcome events in each analysis.	9-10, 23-24
	14b	If done, report the unadjusted association between each candidate predictor and outcome.	21
Model specification	15a	Present the full prediction model to allow predictions for individuals (i.e., all regression coefficients, and model intercept or baseline survival at a given time point).	10
	15b	Explain how to use the prediction model.	9-10
Model performance	16	Report performance measures (with CIs) for the prediction model.	10-11, 22,25
Discussion			
Limitations	18	Discuss any limitations of the study (such as nonrepresentative sample, few events per predictor, missing data).	13-14
Interpretation	19a	For validation, discuss the results with reference to performance in the development data, and any other validation data.	11-12
	19b	Give an overall interpretation of the results, considering objectives, limitations, and results from similar studies, and other relevant evidence.	13-14
Implications	20	Discuss the potential clinical use of the model and implications for future research.	13-14
Other Information			
Supplementary information	21	Provide information about the availability of supplementary resources, such as study protocol, Web calculator, and data sets.	N/A
Funding	22	Provide the source of funding and the role of the funders for the present study.	14

4.) Page 7 Lines 159-169: For all time-varying potential predictors (e.g. mechanical ventilation), please clarify they were obtained before or at time of index culture. If they were past this time point, would exclude from model as would be unavailable for prediction till later in patient encounter.

Response: We have added clarification to when applicable predictor variables were obtained (i.e., prior to, at the time of, or during) relative to index culture collection or index encounter (Page 8, Lines 172-174; 176, 179-182). No variables were collected past the index culture date; therefore, we did not exclude any variables.

- 5.) Page 9 Lines 198-202: Again, as recommended per TRIPOD, would include the full regression model (in addition to current risk score model) including the beta coefficients and intercept.

Response: To add transparency, we have added the full regression model equation to the manuscript (Page 10, Lines 217-220).

- 6.) Page 10 Lines 28-31 and Pages 12-13 Lines 284-290: How was NPV and PPV derived if this was a case control study and therefore prevalence of the pathogen is not derivable from these patients? Do you have the prevalence from other data within your system or was assumed from literature?

Response: As stated by the reviewer, the case-control design precludes us from determining the true prevalence of *Pseudomonas* at our institution. We removed NPV and PPV throughout the manuscript. Due to variable rates of *Pseudomonas* in the literature and possible difference in reported prevalence based on our institutions case-mix index, we feel this further supports removal of NPV and PPV from the manuscript in lieu of assuming a true prevalence for our population using current literature. Page 11, Line 244; Page 12, Line 280; Page 13, Line 295; Page 14, Line 306-307 updated with removal of NPV/PPV related text. Also, NPV and PPV values removed from Table 5.

- 7.) Page 10 Lines 228-231: "the scoring model improved the accuracy" compared to what? What was it improved from? As no comparison was made (e.g. to say clinical judgement), would consider revising verbiage. Similarly, on page 12 Lines 272-277, would avoid "prevalence" verbiage given this is a case control study.

Response: Verbiage updated to reflect that model was accurate in identifying isolation of *Pseudomonas* and not necessarily improved upon (Page 11, Lines 256-257). All affected references to prevalence in the results and discussion sections has been removed (please see item 6 above)

Reviewer #2:

The authors present a single-center retrospective study to identify their center's local risk factors for pneumonia due to *Pseudomonas*. They acknowledge that other risk scores have been developed but do not perform as well outside of the center at which they were derived. To this end, they sought to look at their population to determine risk factors for *Pseudomonas* amongst all those admitted for CAP/HAP/VAP. One strength of the study is the use of the standardized CDC definition of pneumonia and microbiologic diagnosis in all included patients. One weakness, which they acknowledge, is the heterogeneous inclusion of CAP/HCAP/VAP/HAP cases in the score.

1. Line 99: Recommend clarifying this; rather than "4% risk of resistance" perhaps saying "4% chance of developing a resistant organism".

Response: An update to text now reflects reviewer suggested wording (Page 4, Lines 98-99).

2. Line 144: patients selected for the control group were not selected "at random" if every 3rd non-*Pseudomonas* culture was selected

Response: We have added clarity on our process for selecting the control group and removed any reference to random selection (Page 7, Line 151-160).

3. How were COPD, emphysema, bronchiolitis and asthma defined? There is a lot of overlap between these entities. Since COPD was part of the final score, would suggest adding more clarity surrounding how these were defined.

Response: To address definitions of preexisting respiratory conditions, we have added that these were identified via patient problem list or ICD9/10 coding (Page 8, Lines 184-187).

4. Most of the predictive weight in the score comes from a prior positive *Pseudomonas* culture. In practice, empiric *Pseudomonas* coverage would be given to these patients without calculating a risk score. The patients without prior culture are the ones where a prediction score would be useful. Would recommend presenting an alternative model that does NOT include this variable, along with performance characteristics.

Response: We acknowledge that in some clinical practice settings that prior isolation of *Pseudomonas* alone may be sufficient to trigger antimicrobial coverage for this pathogen. Within our model, the prior isolation of *Pseudomonas* variable alone exceeds the score threshold of 11; therefore, we also considered the logistic regression model omitting it. The resulting coefficients were similar except for the long-term steroid use coefficient, which increased from 11 to 12 for a maximum score of 30 and a score of 9 to differentiate between cases and controls. Sensitivity, specificity, accuracy, and model fit were similar (data not shown). We feel that the model with inclusion of the prior cultures is reflective of prior described studies and has been an included variable in these models¹⁻². To address the reviewer's concerns, we have added a statement in the manuscript to reflect that the model with omission of prior cultures was not vastly different from a model inclusive of this variable (Page 10, Lines 234-238 and Page 11, Lines 239-240).

References:

1. Webb BJ, Dascomb K, Stenehjem E, Vikram HR, Agrwal N, Sakata K, Williams K, Bockorny B, Bagavathy K, Mirza S, Metersky M, Dean NC. 2016. Derivation and multicenter validation of the drug resistance in pneumonia clinical prediction score. *Antimicrob Agents Chemother* 60:2652–2663. 10.1128/AAC.03071-15.
2. Oliver MB, Fong K, Certain L, Spivak ES, Timbrook TT. 2021. Validation of a Community-Acquired Pneumonia Score To Improve Empiric Antibiotic Selection at an Academic Medical Center. *Antimicrob Agents Chemother* 65(2):e01482-20. 10.1128/AAC.01482-20.

In the introduction, the authors correctly state that a limitation of prior studies is lack of availability to other sites, as mentioned above. Here, they present a study that can be potentially useful at their own site, and rightly point out that the generalizability needs to be determined.

Response: We appreciate the perspective of the reviewer and hope our study can further guide additional studies to determine the generalizability of our clinical prediction model.

Sincerely,

Meagan Johns, PharmD, MBA, BCCCP, BCPS, BCNSP

May 3, 2022

Dr. Meagan Lee Johns
University of Texas Southwestern Medical Center
Pharmacy
Dallas, Texas

Re: Spectrum00424-22R1 (Derivation and validation of a clinical prediction score to identify the isolation of Pseudomonas in pneumonia)

Dear Dr. Meagan Lee Johns:

Your manuscript has been accepted, and I am forwarding it to the ASM Journals Department for publication. You will be notified when your proofs are ready to be viewed.

Sincerely,

Bonnie Prokesch
Editor, Microbiology Spectrum
